# The Use of Interdisciplinary Approaches to Understand the Biology of *Campylobacter jejuni*

**DOI:** 10.3390/microorganisms10122498

**Published:** 2022-12-16

**Authors:** Paulina A. Dzianach, Francisco J. Pérez-Reche, Norval J. C. Strachan, Ken J. Forbes, Gary A. Dykes

**Affiliations:** 1Geospatial Health and Development, Telethon Kids Institute, Perth 6009, Australia; 2School of Natural and Computing Sciences, University of Aberdeen, Aberdeen AB24 3FX, UK; 3School of Medicine, Medical Sciences and Nutrition, University of Aberdeen, Aberdeen AB24 3FX, UK; 4School of Agriculture and Food Sciences, University of Queensland, Brisbane 4072, Australia

**Keywords:** foodborne pathogens, infection control, biological models, statistical models, mathematical models, multiscale descriptions

## Abstract

*Campylobacter jejuni* is a bacterial pathogen recognised as a major cause of foodborne illness worldwide. While *Campylobacter jejuni* generally does not grow outside its host, it can survive outside of the host long enough to pose a health concern. This review presents an up-to-date description and evaluation of biological, mathematical, and statistical approaches used to understand the behaviour of this foodborne pathogen and suggests future avenues which can be explored. Specifically, the incorporation of mathematical modelling may aid the understanding of *C. jejuni* biofilm formation both outside and inside the host. Predictive studies may be improved by the introduction of more standardised protocols for assessments of disinfection methods and by assessment of novel physical disinfection strategies as well as assessment of the efficiency of plant extracts on *C. jejuni* eradication. A full description of the metabolic pathways of *C. jejuni*, which is needed for the successful application of metabolic models, is yet to be achieved. Finally, a shift from animal models (except for those that are a source of human campylobacteriosis) to human-specific data may be made possible due to recent technological advancements, and this may lead to more accurate predictions of human infections.

## 1. Introduction

Bacterial infection through the ingestion of contaminated food is a major cause of death and illness around the globe [1]. *Campylobacter jejuni* and *Campylobacter coli* are bacteria frequently found in the intestinal microbiota of farm birds and other domesticated animals such as pigs, cattle, or sheep [2,3]. Although *C. jejuni* generally does not pose a serious threat to healthy individuals, The World Health Organisation (WHO) considers *Campylobacter* to be in the top four causes of diarrheal disease, and furthermore, the most common bacterial cause of human gastroenteritis in the world [1]. In 2020, the American Centers for Disease Control and Prevention (CDC) reported that confirmed *Campylobacter* infections rose by 13% in 2019 in comparison to the 2016–2018 baseline [4]. On the other hand, the European Food Safety Authority (EFSA) reported a stable trend of campylobacteriosis during 2015–2019, with 220,682 confirmed cases in 2019 [5]. These facts illustrate that the *C. jejuni* contamination problem has not been solved, and the urgency to address it is very high.

The poultry industry is recognised as a significant risk for the spread of campylobacteriosis, specifically because *C. jejuni* easily spreads asymptomatically in chicken populations, causing subsequent contamination of water distributed to other farm animals and the contamination of meat intended for human consumption [6]. Broiler chicken has been identified as the most common cause of human infection by *C. jejuni* [3,7,8]. In an Australian study of retail meat samples collected weekly from 2016 to 2018, 85% of chicken samples tested positive for *Campylobacter* [9]. In comparison, beef, lamb, and pork retail samples tested positive for *Campylobacter* in 14%, 38%, and 31% of samples, respectively [9]. These data suggest a need for improvement in pathogen control at the different steps of the meat production process.

Given that *C. jejuni* can only grow at 30–45 °C in a microaerobic atmosphere and its sensitivity to stresses encountered outside of the host, the high prevalence of this bacterium on retail samples is a peculiar phenomenon. It has been hypothesised that the ability of *C. jejuni* to survive outside of the animal host may be attributed to several survival mechanisms such as biofilm formation, entering a viable-but-not-culturable (VBNC) state, or interactions with free-living amoebae [10,11,12].

Due to its fastidious growth requirements and the difficulty in recovering it from the environment [13], the study of this organism in situ is relatively more challenging compared to organisms such as *Salmonella*, *Escherichia coli*, or *Pseudomonas aeruginosa.* The use of mathematical and statistical models or analysis of whole genome reconstructions, for example, may aid research by reducing the number of resources required to expand our knowledge of this pathogen. Specifically, this can be achieved by providing directions to experimental analysis or focusing on the key data that need to be collected.

The studies that have been applied in *C. jejuni* research may be grouped in terms of their resolution (Figure 1) as those that focus on the microorganisms themselves (i.e., biofilm formation studies, predictive studies, metabolic models), the individual host (animal models), or a population of hosts (epidemiological studies). The biofilm formation studies such as those discussed in this review can extrapolate commonly observed patterns, providing a general framework through which *C. jejuni* biofilms can be more easily controlled. Namely, these studies may inform the development of strategies inhibiting the survival of *C. jejuni* colonies in biofilms [14]. Predictive studies have been employed to improve food safety at various stages of production by determining whether *C. jejuni* is capable of growth or survival in given conditions (defined by temperature, pH, flow, etc.) and thus whether *C. jejuni* prevalence may cause a food safety issue in these conditions [15]. Metabolic modelling uncovers complex metabolic pathways and thus also cell–cell interactions [16]. Metabolic models are recognised for their usefulness in the biotechnology field, and they are applied for the design of new drugs and vaccines or the engineering of cells by changing their metabolism [17,18]. In the context of *C. jejuni*, these models have the potential to supplement microorganism-level research (i.e., predictive and biofilm formation studies) through their ability to predict cell physiology at the resolution of a single cell. Furthermore, metabolic models have the potential to help ease the disease burden caused by *C. jejuni* ingestion by their power to identify target proteins for drug or vaccine development [18] or by identifying factors that affect pathogen virulence [19]. This, in turn, could aid *C. jejuni* research on an individual host level. In particular, the identification of metabolic factors affecting virulence or the ability to colonise the host may motivate further case studies in which animal models are employed. These are the models in which animal subjects are used to study the disease in vivo [20]. Finally, assessment of *C. jejuni* incidence and disease data at a host population level through the use of epidemiological models has the potential to identify the most prominent sources of infection [3], factors affecting the severity of illness [21] or risk of post-infection complications [22], among many other useful possibilities.

In the following sections, we present the range of research disciplines mentioned above that were employed to improve our understanding of *C. jejuni*. The schematic diagram of the presented disciplines can be seen in Figure 1.

## 2. Microorganism Level

### 2.1. Biofilm Formation Studies

The structure and composition of a mature biofilm form a physical and chemical barrier that protects bacterial cells from harsh environmental conditions and antimicrobial agents. There is evidence suggesting an increased survival of *C. jejuni* biofilm cultures under adverse conditions as compared to the same type of cells in planktonic cultures [14,23]. For example, although most *C. jejuni* strains are not able to grow in aerobic conditions, it has been shown that in biofilms, they survive significantly longer compared to planktonic cells under the same aerobic atmosphere [23]. The increased length of survival of cells within the biofilm, when exposed to atmospheric conditions, may increase the chance of the bacteria being transferred to a more suitable environment in which it can grow, such as a living host. Furthermore, horizontal gene transfer, which may be particularly enhanced within biofilms due to the proximity of individual cells, has been found to increase the antimicrobial resistance of *C. jejuni* [24]. This evidence suggests that the protective nature of biofilms may allow *C. jejuni* to colonise many different environments and could explain why it is ubiquitous in the agricultural, food, and medical sectors [23].

Biofilm formation by any bacterial species occurs in the following stages: surface attachment, microcolony formation, biofilm maturation, and cell detachment and dispersal [25] (Figure 2). This is the simplest, general biofilm life cycle description. The particular mechanisms that facilitate biofilm formation, however, vary between different species. For example, the composition of the extracellular matrix, mechanisms facilitating surface attachment, or responses to environmental factors, may differ [14]. In order to create a more detailed description of biofilm formation, one must focus on the properties of the species of interest.

An extensive laboratory analysis, which identified the pillars of *C. jejuni* biofilm formation under static conditions, resulted in the development of a general description of *C. jejuni* biofilms [14]. For *C. jejuni*, adhesion is believed to be facilitated by flagella since aflagellate mutant strains have an impaired ability to attach to surfaces [14,23,26,27] unless the surface conditions are particularly favourable [28]. The study of Svensson et al. [14] additionally revealed an association of *C. jejuni* biofilm maturation with bacterial lysis, which was later confirmed in another study [29]. Confocal microscopy imaging of *C. jejuni* biofilms showed an abundance of eDNA present in mature biofilms, which has also been confirmed in another study, where additionally lipids, proteins, and polysaccharides were reported as other key constituents of the extracellular matrix [29]. Biofilm formation was significantly reduced in the presence of DNAse I; however, no significant difference in surface attachment was observed, indicating that eDNA is not necessary for the attachment of *C. jejuni* to surfaces. On the other hand, in the conditions for which eDNA release and biofilm formation were enhanced (MHB with sodium deoxycholate), horizontal gene transfer, manifesting through the recovery of colonies exhibiting combined antibiotic resistance of two parental strains that initiated the formation of the biofilm, was found to be increased. This property has also been later confirmed in another study [30]. A replica of the model built on the collection of experimental evidence gathered by Svensson et al. [14] on *C. jejuni* biofilms can be found in Figure 3. In summary, the study concluded that the biofilm formation of *C. jejuni* may be triggered by adverse environmental conditions, and initial attachment is facilitated by flagella. Furthermore, as the biofilm matures, an abundance of eDNA is released, with evidence suggesting that this release is in significant part due to a lytic process. Finally, the study presented evidence of increased stress tolerance and horizontal gene exchange in well-formed biofilms, which exhibited an abundance of eDNA [14].

The ability of *C. jejuni* to successfully integrate eDNA into its existing genome has been suggested to account for the apparent genetic variation between *C. jejuni* strains [31]. The process of horizontal gene transfer, specifically the binding of double-helix DNA strands onto the bacterial surface, followed by degradation of one of the strands into nucleotides and the integration of the other strand into the genome of the bacterial host, begs the question as to whether the nucleotides released could be utilised as a nutrient source for *C. jejuni*. The use of eDNA as a source of carbon, nitrogen, and phosphate in nutrient-limiting conditions has been confirmed for many species of bacteria [32]. Further research is required to establish if this mechanism is also relevant to *C. jejuni.* A recent study of the genome of several *C. jejuni* strains suggest that this could be the case since genes were identified for nucleotide metabolism and transport [33].

Apart from the general properties of the structure, composition, and physiology of *C. jejuni* biofilms, the relationship between *C. jejuni* biofilm formation and environmental conditions has also received a considerable amount of interest. Whether aerobic conditions enhance or inhibit biofilm formation is not clear yet. In some studies, microaerobic conditions have been found to produce higher amounts of biofilms [26,29], while in others, the opposite was the case [34]. This may be partly attributed to specific properties of the media and particular strains, as a systematic study comparing biofilm formation in various media (MHB, Bolton, and Brucella broths) and eight strains revealed all possible types of effects of cultivation in aerobic conditions on biofilm formation (i.e., biofilm formation inhibited, enhanced or equivalent), which seemed to depend on the type of media used [35]. Specifically, it was suggested that the presence of sodium bisulphite, which is an agent reducing levels of dissolved oxygen (DO) in Bolton and Brucella broths, may have played a role in biofilm formation being equivalent or higher in aerobic conditions compared to microaerobic conditions in these media. In contrast, in Mueller Hinton Broth, which lacks oxygen-reducing ingredients, biofilm formation was either equivalent or lower in aerobic conditions [35]. Apart from the effect of atmospheric oxygen on the biofilm-forming ability of *C. jejuni*, there have also been some confounding postulates made on the effect of nutrient levels on the biofilm formation of *C. jejuni*. Again, further research is needed to better understand these effects on biofilm formation. Some studies suggested that lower nutrient media may promote biofilm formation of *C. jejuni* through comparisons of biofilm formation in nutrient-low MHB with higher nutrient NB2 [36] Bolton or Brucella Broths [26]. On the other hand, another study found that higher planktonic growth was obtained in a highly nutritious Tryptone Soya broth with 0.6% yeast extract (TSBYE) compared to NB2 and MHB [37]. These observations beg to question as to how *C. jejuni* biofilms would perform when cultured in TSBYE, compared to MHB.

The qualitative biofilm formation models described in this section aided in building a general picture of *Campylobacter jejuni* biofilms by finding common traits observed in biofilm assays of this species (e.g., eDNA as a major component of biofilms and lysis as an important process involved in biofilm formation, natural ability for horizontal gene transfer, flagella as an important structural component initiating surface attachment, etc.). These common traits might be used to inform the development of control measures for *Campylobacter jejuni* contamination through biofilms, which would be potentially applicable to a wide range of *C. jejuni* strains. For example, since the results of biofilm matrix composition studies have indicated that eDNA is a major component of *C. jejuni* biofilms, researchers have turned to studying the effect of *C. jejuni* biofilm treatment with a DNA disruptive enzyme (DNAse), which has shown to result in disruption of biofilms in several *C. jejuni* strains [29,30,38].

Apart from the application of DNAse in biofilm control, other research avenues may stem from a general description of *Campylobacter jejuni* biofilms. For instance, such descriptions may lead to the development of mathematical models of *C. jejuni* biofilms. Biofilm modelling using mathematical descriptions has already proven to be useful in answering particular questions related to areas in which biofilm formation is important, such as wastewater management or the food and medical sectors [39]. Such models commonly include computer simulations, which allow for testing hypotheses related to an occurrence of observed phenomena or for the prediction of biofilm formation for a wide range of possible scenarios [39]. However, as far as we know, a specific mathematical model of *C. jejuni* biofilms has not yet been reported. From this section, it is evident that there are many unanswered questions regarding the general properties of *C. jejuni* biofilms—for instance, there is a need for a clearer understanding of the effect of environmental (nutritional and atmospheric) conditions on *C. jejuni* biofilm formation and its survival capabilities. Mathematical models can simulate a much larger set of conditions than might be feasible using experimental methods alone. Following this, integrating experimental observation and mathematical models offers unique avenues to uncover biological trends [39].

Mathematical models can only capture certain aspects of any biological system. The extent to which this represents a limitation depends on the research question to be addressed. Nevertheless, the synergy between experimental observations and mathematical models can be exploited iteratively to answer complex research questions. More specifically, experimental observations can be used to formulate an appropriate mathematical model. This model can then be used to make new predictions that will, in turn, motivate additional experiments to test the predictions. This will typically suggest improvements to the model, and the cycle of alternating experiments and models can repeat until we are satisfied with the answers to the research question. This requires fluent communication between experimentalists and mathematical modellers to ensure that, for instance, mathematical models are designed in such a way that their predicted outputs can be verified with observed data. In the case of the survival of *C. jejuni* biofilms, a natural starting point could be a mathematical model for a genetically homogeneous population to explore the effect of environmental conditions. This could then be extended to incorporate the considerable degree of heterogeneity within the *C. jejuni* species to study, for instance, how this may influence the survival in the presence of antimicrobial agents.

### 2.2. Predictive Studies for C. jejuni Survival

Hazard analysis and critical control point (HACCP) principles are considered a cornerstone on which preventative strategies at all stages of food production are developed to ensure food safety [40]. Predictive studies are constituents in the process of following the HACCP principles through the assessment of the efficiency of interventions introduced at the slaughter and food processing stages, which aim to reduce pathogen incidence on food products [40]. In particular, these studies aim to assess how a variable (such as temperature, or a concentration of a biocide, for example) affects the observed reduction in the treated bacterial population.

In the case of *C. jejuni*, such predictive studies generally focus on its elimination from chicken carcasses [40]. These decontamination interventions can be grouped into three categories: physical interventions (hot water, steam irradiation, ultrasound, ultraviolet light, air chilling, freezing, etc.), chemical interventions (organic acids, chlorine, hypochlorite, electrolysed oxidising water and ozonated water, etc.) and biological interventions (bacteriophages) [40].

A 2018 study that compared the effectiveness of several chemical interventions on the reduction in *Campylobacter* and *Salmonella* incidence on chicken carcasses in a post-chill decontamination tank reported peracetic acid (PAA) and cetylpyridinium chloride (CPC) as the most effective methods compared to all other interventions considered, while chlorine and acidified sodium chlorite were found to be the least effective interventions [41]. Although the use of chemicals may be efficient in reducing microbial counts, there are concerns regarding the consumer and environmental safety of the application of chemicals on food products. While in the USA, many chemical decontamination methods are allowed, in the European Union, only lactic acid of up to 5% has been so far approved for use [42]. Instead, physical treatments involving temperature or water are mostly applied in that geographical area [43]. For example, a recent study proposed a promising method for the treatment of carcasses with steam at 95 °C and 120 °C for 3–5 s [42]. Although complete elimination of pathogens may be achieved with steam if applied for long enough, application for 10 s has been previously shown to reduce the quality of meat.

Apart from the application of predictive studies to assess microbial counts along with other indicators of meat quality for discrete values of a given variable (e.g., temperature or time of treatment), attempts have been made to translate empirical observations into theoretical predictions for a wider range of conditions.

One of the existing examples of such studies relevant to *C. jejuni* is an empirical model built to predict the survival of *C. jejuni* as a function of temperature ranging from 4 °C to 30 °C [44]. The authors used a simplified version of the Davey model to describe the temperature dependence of the initial lag time, LT, during which the population size remains approximately constant:(1)$$LT=A+\frac{B}{T}+\frac{C}{T^{2}}$$

Here, *A*, *B*, and *C* are constants to be determined by fitting them to experimental data. For the relationship between the specific death rate (SDR) of the organisms and temperature, the Boltzmann sigmoidal function was found to be a good fit to the obtained measurements [44]:(2)$$SDR=SDR_{\min}+\frac{SDR_{\max}-SDR_{\min}}{1+\exp\left[(T_{50}-T\right)/slope]}$$

Here, $SDR_{\min}$ and $SDR_{\max}$ are the minimum and maximum death rates, respectively, $T_{50}$ is the temperature at which *SDR* is halfway between its minimum and maximum values, and *slope* is the rate of change of *SDR* as a function of temperature between its extreme values.

The results of the study suggested that the lag time decreases monotonically with temperature. Furthermore, specific death rates during the log phase were found to increase with temperature, and this increase occurred at a certain threshold (at the observed range, the threshold appeared to occur between 16 °C and 20 °C). Finally, the study found that the maximum reduction in log CFU/mL of *C. jejuni* organisms on poultry patties or broth was not affected by temperature in the assessed range, i.e., 4–30 °C [44].

While the above study focused on understanding the relationship between temperature and death rate or temperature and lag time, other studies may aim to quantify the relationship between microbial counts as a function of time, given a set of external conditions. With regard to these studies, distinct survival curves as functions of time have been classified and assigned a suitable model distribution [45]. One of the common distributions, which has been found to provide a good fit for some of the types of survival curves, is the Weibull distribution [46,47]. This is a two-parameter distribution with many applications, from survival analysis of live organisms to weather forecasting. In particular, given an initial number of *C. jejuni* cells, $N_{0}$, the number N of surviving cells at time t has been described in terms of the Weibull cumulative distribution function as follows [46]:(3)$$\log N=\log N_{0}-{\left(\frac{t}{\delta }\right)}^{p}$$

Here, *p* is the parameter determining the shape of the distribution (concave or convex), and δ is a scale parameter corresponding to the time for the first 1 − log reduction (simply because $\log N=\log N_{0}-1$ for t=δ).

A striking difference between the two studies described above [44,46] was that although the same medium and temperatures were analysed in both of these studies, in the first one, a considerable lag time was observed (i.e., the population size remained approximately constant in the first days of incubation), while in the second study, the initial reduction in cell numbers was most abrupt and decreased as the time passed. What was consistent in both studies was the emergence of the subpopulation resistant to the conditions they were exposed to, which manifested through the levelling out of the death curve as time progressed. The specific factors responsible for the existence of lag time are not entirely clear. However, reviewing survival curves obtained in various studies reveals that whether or not a lag phase is observed in a given time frame may depend on both the strain tested and the incubation conditions (e.g., temperature, pH, or atmosphere) [48,49,50]. The fact that the two studies described above performed their analysis on different strains of *C. jejuni* may explain the difference in the shapes of the survival curves against time [44,46].

Apart from predictions of *C. jejuni* cell counts on food under various storage conditions, the data obtained from the analysis of survival of *C. jejuni* under various temperatures can be used in predicting the change in *C. jejuni* counts resulting from other food processing practices, e.g., scalding [15], which is a treatment of meat carcass with hot water or steam. A mathematical model incorporating scalding process factors such as the volume of the scalding tank, average contamination of carcass before the scalding, rate of carcasses entering the tank, or the rate of the detachment of the bacteria from the carcass into the scalding water, plus the thermal inactivation data of *C. jejuni* strains subjected to scalding temperatures at varying pH values, could be a useful tool for food processing industries in the analysis of the effect of various factors on contamination of the final product [15]. This particular model predicted that for a relevant range of model parameters, the level of *C. jejuni* contamination in the scalding water achieved a steady state in a short time, suggesting that the scalding process may be one of the sources of cross-contamination in meat processing [15].

In summary, predictive studies are a necessary tool in choosing the right set of decontamination methods at various food processing stages. Unfortunately, there seems to be no silver bullet solution that could lead to a complete eradication of *C. jejuni* contamination of food products. Rather, multifaceted approaches for pathogen control at every step of the production process (“from farm to fork”) need to be further improved [51]. It has been previously recommended that more standardised protocols should be developed for better comparability of results reporting on microbial reductions following a given intervention [40]. Furthermore, it has been recommended, based on current consumer trends and growing environmental concerns, that the assessment of natural disinfection methods (e.g., use of plant-based extracts) might be worthwhile, and some extracts from fruits and seeds have exhibited the potential to reduce the viability of *Campylobacter* on chicken samples without negatively affecting the sensory analysis of the meat [52]. Such methods of chemical decontamination may aid physical decontamination methods with the additional benefit of being easier to accept by legislators and consumers. Moreover, novel physical disinfection methods, such as oscillating magnetic fields, use of enzymes, manothermosonication, pulsed electric fields, etc., may be of interest to consider [40].

### 2.3. Metabolic Modelling and Growth Requirements

Genome-scale metabolic models (GSMs) aim to predict the physiology and metabolism of organisms subjected to given environmental conditions. The development of a metabolic model typically follows four major steps, namely initial metabolic network reconstruction from gene annotation, refining the initial reconstruction with the use of other relevant data obtained from the literature, conversion into a mathematical model, and validation of the reconstruction coupled with further refinement through comparison of the output of the model with reported phenomena [53]. This type of model has been so far utilised more extensively for organisms such as *E. coli* [54] or *P. aeruginosa* [55]. The first metabolic model of *C. jejuni* was proposed by Metris et al. [16]. This model is based on genome sequence data obtained from the NCTC11168 strain and relevant information found in the literature on *C. jejuni*. Where information on *C. jejuni* was lacking, assumptions were made based on the data found for a closely related bacterium species, *Helicobacter pylori.* Information such as reactions for amino acid metabolism and nucleotide metabolism were drawn from the genome annotation. On the other hand, central metabolism reactions were mainly drawn from other literature sources [16]. The model predicted, among other things, the predominance of essential genes associated with aromatic amino acid metabolism, tRNA metabolism and protein synthesis, the TCA cycle, the cell envelope, and purine and pyrimidine metabolism [16].

More recently, another metabolic model has been proposed for the *C. jejuni* M1cam strain, in which specific auxotrophisms of this strain were identified [56]. In particular, the study reported that the M1cam strain is auxotrophic for methionine, niacinamide, and pantothenate. They also found that this strain can produce energy, but not biomass, in the absence of oxygen. By using this metabolic model, the authors were able to design a growth-enhancing media for *C. jejuni* M1cam, which supported a 1.75-fold higher growth rate than that measured for culturing the strain in the Brucella broth, which is a commonly used substrate for *C. jejuni* in laboratory experiments. The design of growth-enhancing substrates for *C. jejuni* may be of interest to anyone engaging in laboratory assays for this species, particularly because it is known for being difficult to grow in the laboratory setting. By uncovering specific metabolic requirements of the organism, metabolic models allow for the substrate design to target these requirements with high precision [56].

The incorporation of metabolic reconstructions into mathematical models of bacterial populations has not yet been reported for *Campylobacter*, although it has been found to produce novel insights about colonies of other organisms such as *E. coli* [54]. The lack of such models for *C. jejuni* may be due to this organism being understudied compared to *E. coli*. Producing a model of such substantial detail [54] requires many organism-specific parameters to be derived from the literature. These include the key metabolic requirements and products, with possible cross-feeding mechanisms, or the rates of compound uptake and growth [54].

Similar to other chemoorganotrophic prokaryotes, *C. jejuni* uses organic compounds as energy sources. In particular, amino acids have been identified as the primary substrate for *C. jejuni* [57]. Although fucose, an abundant sugar in the mammalian gut [20], may sustain *C. jejuni* growth for some strains [58], *C. jejuni* is known to have limited capability of utilising other carbohydrates as substrate [13,59].

A genomic study of three strains of *C. jejuni* identified 486 genes that are essential for *C. jejuni* fitness (its survival and growth). Among these, genes responsible for the metabolism of lipids, coenzymes, carbohydrates, nucleotides, and amino acids were found [33]. The appearance of nucleotides on this list may be particularly interesting when coupled with the findings presented in the previous sections, namely that eDNA forms a major component of *C. jejuni* biofilms [14]. The presence of a nucleotide metabolism pathway suggests that it may be possible for *C. jejuni* to utilise the eDNA released by other cells as a nutrient source, and as there is an abundance of eDNA in *C. jejuni* biofilms, this could potentially be a factor in the survival of *C. jejuni* populations in biofilms.

## 3. Animal Infection Model Level

Animal models can be used to identify virulence factors in *C. jejuni*, determine host responses to the presence of the pathogen, or test the viability of potential treatment methods [60]. The use of non-human primate models, on the one hand, desirable due to their closeness to humans, is limited due to ethical considerations and the difficulty in keeping these animals, among other limitations [60].

Human volunteer studies have also been employed. In one such study directly related to *Campylobacter jejuni*, results suggested that the severity of acquired illness is strain-dependent, the likelihood of exhibiting infection symptoms is dose-dependent, and repeated exposure to a specific strain may increase the immunity of the host [61]. The latter finding agrees with the apparent decrease in colonisation symptoms to *Campylobacter* exposure of people living in developing countries, compared to those living in industrialised countries [60]. In a study using a ferret model, it was shown that the NCTC 11168 *C. jejuni* strain has a low virulence compared with the strain 81–176. Even at high doses, NCTC 11168 caused disease in only one out of nine animals, while all tested animals experienced infection symptoms after administering a high dose of 81–176. Furthermore, the study found a reduction in virulence of strains 81–176 with introduced mutations to their plasmid genes, suggesting that plasmids may be a significant factor in 81–176 virulence [62].

In order to produce an infection model that is relevant to human hosts while maintaining ethical standards, antibiotic treatment of mice, used to eradicate their natural microflora, followed by introducing human microflora into their intestines, has been used [63]. This microflora manipulation resulted in a significant change in the outcome of *C. jejuni* colonisation. Namely, mice with murine microflora were clear of the pathogen after 2 days of infection, while the mice with human microflora were found to be colonised for 6 weeks. The study concluded that specific gut microflora is essential in determining the outcome of pathogen invasion, as the natural murine microflora exhibited resistance to *C. jejuni*, while the immune response of the mice with human microflora mimicked that of human campylobacteriosis [63]. In another mice model study, which used antibiotic treatment prior to infection with *C. jejuni*, it was found that mice fed with a zinc-deficient diet exhibited significantly more severe symptoms of campylobacteriosis than those on a standard or protein-deficient diet. Namely, the mice on the zinc-deficient diet suffered from bloody diarrhoea and exhibited significantly increased weight loss due to the infection in comparison to mice on the other diets, for which only mild symptoms were observed [20].

In recent years, insect models, for example, *Galleria mellonella* infection models, have been used to study various microorganisms as an alternative to mammalian or avian models. Models of this type are desirable due to, for example, reduced costs, improved commercial availability, and lack of ethical approval required for the use of these insects for research [64]. Although insects lack an adaptive immune response, their innate immune response is very close to that of vertebrates [65]. In contrast to mammalian or avian hosts, which are usually infected orally [20,61,63], the insect larvae may easily be directly injected with a specific dose of the studied pathogen. As a result, more direct comparisons of virulence between strains may be derived [64]. Typically, an intrahemocoelic injection method is used for inoculation of the larvae, and it is recommended that 10–20 larvae are used for each tested condition for statistical significance [65]. Markers of the disease include melanisation, a decrease in cocoon formation or motility, and death [65].

In one such study using a *G. mellonella* as a model organism for testing *C. jejuni* virulence, the effect of larvae infection with 67 *C. jejuni* isolates was tested [66]. In congruence with common practice, a fixed inoculum size was directly injected into the haemocoel, and the larvae were incubated at 37 °C before assessment. One of the interesting observations was that *C. jejuni* cells recovered from infected larvae haemolymph were found to be in a coccoid rather than the characteristic spiral shape. Furthermore, when infecting cultured mammalian and insect macrophages with *C. jejuni*, cell numbers were found to drop 100–1000-fold in comparison to the initial inoculum size in the first 4 h post-infection and then remain constant or increase again when counted at the 24 h mark [66]. This finding suggests that *C. jejuni* cells experience stress at the initial stages of infection, but the population as a whole may be able to overcome it at later stages, provided that the initial inoculum is of sufficient size. Finally, from the comparison between larvae survival after a challenge with six different MLST types, it was suggested that the ST-21 group exhibits the least virulence (with the mean survival rate at ~95%), and the highest virulence was observed for the group ST-257 (mean survival rate at ~76%). In contrast to the findings of the study outlined above, another *G. mellonella* study revealed a high virulence of a *C. jejuni* poultry isolate 13126, which belongs to the ST-21 clonal complex [67]. Although this particular isolate was not considered in the previously described study [66], this result calls for caution to be exercised before making general postulates about differences between the virulence of MLST groups.

Apart from generating particular data indicating the relative virulence of strains or properties of the host, which may influence the severity of disease symptoms, important general theories have also been developed from this class of research. Data obtained from infection studies have led to the development of a Beta Poisson dose–response model [68], intending to predict the probability of infection or illness based on the administered dose. The Beta Poisson model has paved the way for future dose–response models and has found applications for a wide range of pathogens beyond *Campylobacter* species [69,70,71,72].

Although animal models have provided a plethora of information on many diseases, the variations between species have been reported to be a huge limitation, as the predictions of disease and effectiveness or side effects of tested treatments do not necessarily translate well from one species to another [73]. *C. jejuni* is a good example of this, as it is believed that apart from some exceptions, *C. jejuni* does not generally cause illness in its other common hosts, while many cases of human disease caused by *C. jejuni* are reported each year [74]. It has been suggested that modern technology may allow the shift from animal models to human-relevant data by in vitro analysis of the effect of disease on human tissues or genomics approaches that may identify disease-specific genetic markers, for example [73]. It has been indicated that increased accessibility to human tissues of patients and healthy individuals for research purposes is essential to achieve statistically relevant results [73]. In the case of *C. jejuni*, studies of human and poultry infection patterns may be of most interest.

## 4. Epidemiological Studies at the Host Population Level

Epidemiology is a branch of research dedicated to finding the causes, risk factors, and transmission pathways associated with an illness, as well as predicting the impact of the disease on the population and developing suggestions for optimal control measures [75]. Epidemiology studies rely on the analysis of real-life data associated with a given disease (e.g., data collected from clinical records) [75]. Epidemiology models are an important component of public health research, and as such, many such models have been developed for analysing data relevant to *C. jejuni* to minimise its burden on populations worldwide.

Since epidemiology models rely heavily on data, statistical procedures are at the forefront of these types of studies, especially case–control studies, which aim to identify and quantify risk factors associated with a disease. For example, case–control studies may quantify the relationship between a dependent variable, such as disease incidence, and independent variables, such as geographical location, age, gender, etc. Multivariate logistic regression models have been used in particular in studying these relationships [76,77]. It has been identified that contact with contaminated or undercooked retail chicken, international travel, eating in a restaurant, direct contact with animals, and climate conditions are among the significant risk factors for acquiring a *C. jejuni* infection [74]. Furthermore, risk factors associated with the susceptibility of individuals may be uncovered with case–control studies, e.g., the use of proton pump inhibitors has been associated with increased symptomatic *C. jejuni* infection rate [78]. Other case–control studies identified evidence for an increased risk of campylobacteriosis patients developing irritable bowel syndrome (IBS) [22], functional dyspepsia (FD) symptoms [79], or celiac disease [80]. Extra gastrointestinal post-infection complications associated with *C. jejuni* include Guillain–Barre syndrome, Miller–Fisher syndrome, bacteraemia, septicaemia, cardiovascular complications, meningitis, reactive arthritis, and reproductive system failures [81].

Source attribution studies also aid epidemiology research by examining relative proportions of cases attributable to different sources [82]. Multilocus sequence typing (MLST) allows the phylogeny of isolates to be traced, which has contributed to the finding that chicken is the most prominent source of human *C. jejuni* infections, followed by cattle and sheep [3]. A recent analysis using whole genome sequences led to similar conclusions [83]. Furthermore, MLST has helped to classify *C. jejuni* isolates into distinct, highly diverse lineages, which aids in explaining the observed variation in *C. jejuni* phenotypes for different strains or strain variations. This categorisation of isolates in terms of their phylogeny is a key component in Genome-Wide Association Studies (GWAS), where specific genetic factors associated with a given phenotype can be uncovered [84]. For example, in 2017, a GWAS study found lineage-specific genetic factors that may influence the clinical incidence of *C. jejuni* [85]. Interestingly, among the genes which were found to be associated with the increased clinical incidence of *C. jejuni* in the ST-21 clonal complex, *kpsC*, and *kpsD* genes, which are believed to contribute to surface adhesion and biofilm formation, were identified.

Another key area in which the GWAS studies are employed is the surveillance of antimicrobial resistance. Antimicrobial or drug resistance is a prevalent problem when dealing with any pathogen, as it affects the efficacy of treatment, and the resistance status of the pathogens may change over time through mutations or natural selection. Surveillance of antimicrobial resistance may facilitate improved control over any given pathogen, as it allows making informed decisions on which antimicrobial treatment is best suited in a given context, and it leads to the development of strategies designed to limit the spread of resistant genotypes. For *C. jejuni* specifically, whole genome sequencing has been applied to identify genotypes resistant to specific antimicrobials. Strong correlations have been confirmed between the resistant phenotypes and genotypes in several studies [86,87,88], indicating that genome sequencing may be a reliable tool for monitoring the antimicrobial resistance of *C. jejuni.* Overall, although bacterial GWAS is not exempt from challenges [84], the advent of inexpensive next-generation sequencing technology, together with the development of advanced statistical methods [89], make GWAS a promising framework for identifying the genetic basis of bacterial phenotypic traits.

Lastly, quantitative risk assessment (QRA) methods, among other uses, may employ regression epidemiology models to identify acceptable thresholds for a value of a given risk factor [90]. For *Campylobacter jejuni*, for instance, quantitative risk assessment methods were applied to analyse the prevalence of this pathogen at various stages of food processing and thus pointed to specific areas in the process which may need improvement [91]. With the use of the QRA approach, it has also been recently suggested that a total eradication of *C. jejuni* on retail products may not be necessary, as only highly contaminated products pose a significant risk to consumers [74]. Although a number of transmission pathways of *C. jejuni* to humans have been described, it is believed that there are still some which are yet to be discovered. Apart from discovering the ways humans may come into contact with *C. jejuni*, it has been suggested that focusing intervention strategies at the source (i.e., the farm) could subsequently lead to a decrease in *C. jejuni* prevalence across both known and unknown pathways [74]. In this space, a recent study by Rawson et al. [92] suggested that the infection susceptibility of individual birds is a key factor influencing the spread of *Campylobacter* among chicken flocks. The study implied that the relatively higher frequencies of some strains within the flock were more influenced by these strains being initially ingested by particularly susceptible birds rather than by phenotypic differences between the strains—these highly susceptible birds would then shed the ingested strains in large quantities, causing contamination of the rest of the flock. The conclusion was that the health and welfare of individual birds should be considered to reduce *C. jejuni* colonisation on the farms, given that the immune responses of the chickens have been shown to be impacted by welfare measures. One possible limitation of implementing control strategies on the farm level is that due to *C. jejuni* being generally safe for livestock, farmers may lack the incentive to invest in measures designed to limit *C. jejuni* colonisation in their flocks. The introduction of rewards (e.g., quality certifications) or policies may increase the incentive for farmers to implement more protective measures. A recent example of such an incentive is the ban on thinning procedures on RSPCA-approved farms introduced in 2016, following a report released by the European Food Safety Authority, which linked thinning to increased *C. jejuni* colonisation among broiler chickens [43]. Apart from introducing more control of *C. jejuni* on the farms, it has been suggested that finding a threshold for an acceptable level of meat contamination at the end of the processing stage and discarding or cooking highly contaminated samples may decrease the burden of *C. jejuni* on the health of populations worldwide [74]. Furthermore, close monitoring of new findings achieved by predictive models, which may indicate novel disinfection strategies, may also aid epidemiology studies by motivating the assessment of these strategies on a larger scale, which may, in turn, lead to policy changes and improvement in control of *C. jejuni* transmission.

## 5. Summary

We presented a range of disciplines that have been applied to understand and control *Campylobacter jejuni* and described recommendations for future research in these areas. Specifically, the incorporation of mathematical modelling may aid the understanding of *C. jejuni* biofilm formation both outside and inside the host. Predictive studies may be improved by the introduction of more standardised protocols for assessments of disinfection methods and by assessment of novel physical disinfection strategies as well as assessment of the efficiency of plant extracts on *C. jejuni* eradication. A full description of the metabolic pathways of *C. jejuni*, which is needed for the successful application of metabolic models, is yet to be achieved. A shift from animal models (except for those which are a source of human campylobacteriosis) to human-specific data may be made possible due to recent technological advancements, and this may lead to more accurate predictions of human infections. Epidemiology models may be aided by the inclusion of clear instructions regarding the prescribed usage of statistical approaches in the documentation of generally used statistical software packages, as their misapplication has been reported to be of concern [93]. Furthermore, monitoring advancements and potential pathogen control strategies may motivate testing their efficiency on a larger scale through epidemiological studies, which in turn may lead to improved control over *C. jejuni* globally.

In this review, we tried to make it clear that a combination of different techniques and focus on various aspects, from a scale of the genome, through bacterial communities, up to affected host populations, are all important pieces of the health challenge puzzle posed by *C. jejuni.* Taken together, the proposed advancements could ultimately facilitate the reduction of *C. jejuni* burden on public health.

## Figures and Tables

**Figure 1 microorganisms-10-02498-f001:**
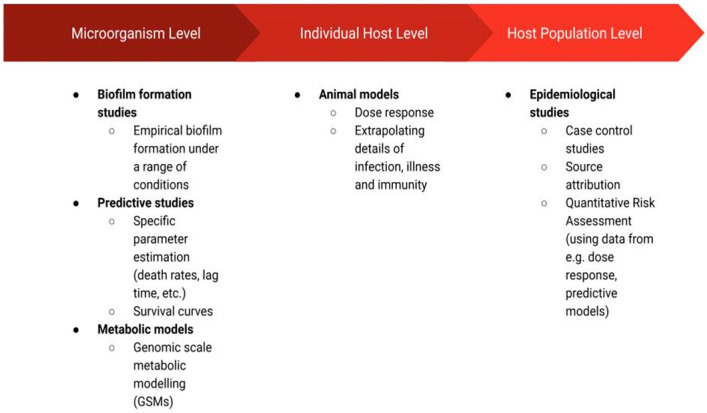
Schematic diagram of the disciplines discussed in this review, which were employed to study *Campylobacter jejuni* species.

**Figure 2 microorganisms-10-02498-f002:**
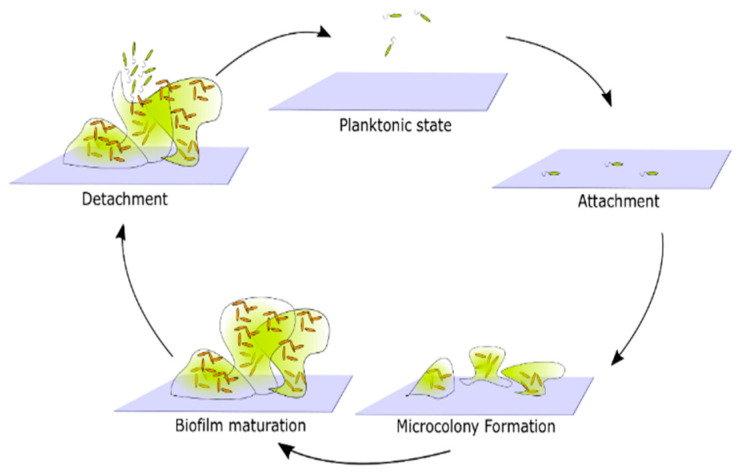
Generic representation of the life cycle of biofilms. Cells in a planktonic state migrate and attach to the surface. Attached cells form microcolonies by reproduction and generation of extracellular products that together form a biofilm matrix. Over time microcolonies begin to merge, and a mature biofilm emerges. Eventually, some cells detach from the biofilm and return to the planktonic state.

**Figure 3 microorganisms-10-02498-f003:**
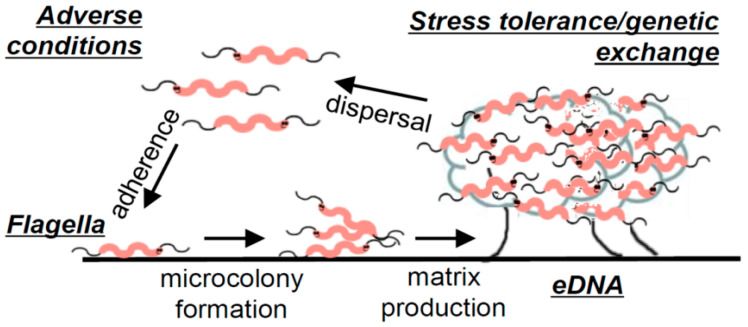
Illustration of *C. jejuni* biofilm formation. Figure taken from “Flagella mediated adhesion and extracellular DNA release contribute to biofilm formation and stress tolerance of *Campylobacter jejuni*” by Svennson et al. [14] (reprinted under an open access license).

## Data Availability

Not applicable.

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
