# Peer review of "The Use of Interdisciplinary Approaches to Understand the Biology of Campylobacter jejuni"

_microorganisms, 2022, doi:10.3390/microorganisms10122498_

Round 1

Reviewer 1 Report

The authors reviewed, described  and evaluated biological, mathematical, and statistical approaches used to understand the behaviour of Campylobacter jejuni, a foodborne pathogen and suggested future avenues which can be explored. I really enjoyed reading the review.

See below for my concerns and minor corrections that would make the paper appealing to wider readers.

Metabolic modelling and growth requirement

Can the author include DOI: 10.3389/fmicb.2020.01072 as a reference. Although the study specifically looked at C. jejuni M1, the metabolic model can be applied to any other C. jejuni with minor modifications.

Microorganism level

I understand the authors want to focus on biofilm as a survival mechanism, but it is important to mention other mechanisms. Can the authors also include other modes of survival and persistence in the environment in addition to biofilms such as VBNCs. It is known that this contributes to C. jejuni persistence. I also suggest the authors include other survival mechanisms of C. jejuni such as interactions with free-living amoebae, both of these are relevant. A paragraph in the introduction would suffice. Can the authors also include limitationsof the mathematical modelling.

Can the authors comment on the heterogeneity of C. jejuni strains and how this could be a challenge to tackle using mathematical modelling.

Author Response

  1. Can the author include DOI: 10.3389/fmicb.2020.01072 as a reference. Although the study specifically looked at C. jejuni M1, the metabolic model can be applied to any other C. jejuni with minor modifications.

Thank you for pointing us towards this article. It is indeed an extremely useful one, marking a massive achievement in the Campylobacter field. We were more than happy to include it in the metabolic modelling section of the manuscript.

  1. I understand the authors want to focus on biofilm as a survival mechanism, but it is important to mention other mechanisms. Can the authors also include other modes of survival and persistence in the environment in addition to biofilms such as VBNCs. It is known that this contributes to C. jejuni persistence. I also suggest the authors include other survival mechanisms of C. jejuni such as interactions with free-living amoebae, both of these are relevant. A paragraph in the introduction would suffice.

Thank you also for this comment. We agree that it is important to mention these survival mechanisms, as both of them are highly relevant to improve control over C. jejuni. We added a paragraph in the introduction referencing to these survival mechanisms, as requested.

  1. Can the authors also include limitations of the mathematical modelling. Can the authors comment on the heterogeneity of C. jejuni strains and how this could be a challenge to tackle using mathematical modelling.

Limitations of mathematical modelling were added, including the heterogeneity of C. jejuni strains.

Reviewer 2 Report

This review  suggests   new possible approaches aimed at the implementation of biosecurity measures against Campylobacter , considering the fact  there are no successful control  plans against it. The limit of this approach, already proposed in other contest, is represented by the difficulty of standardization  as there are many variables  affecting  the characteristics of the bacterium, from the physical to the metabolic and pathogenic ability , as well as the farming conditions of birds . However if  associated  to others measures  could be useful in application of corrective actions applied in the meat chicken production  chain addressed to minimize prevalence Campylobacter .   The authors did not refer to the phenotypic and genotypic characteristics of antibiotic resistance of Campylobacter often related to morphological characteristics, i.e biofilm, in some microorganisms . Nowadays, the antimicrobial resistance of Campylobacter spp. is considered more “insidious”  than  the its effective zoonotic aspect .  Finally  you should consider, if you like,  an interesting work of Rawson et al (2020) as possible reference, for your introduction or discussion .

Please note  that  minor errors are in the text ;

Use italics for Campylobacter  jejuni fig 1. Fig 3 line 268

Line 172-173 I would omit this sentence, it is generally used when the Authors are directly responsible for the study.

 Line 231-232 “ Instead, physical treatments involving temperature or water are mostly applied in that region”. Which region are you referring to?  ? Please clarify sentence

line 398  per os or orally 

Reference 

10.3389/fmicb.2020.576646

Author Response

  1. This review suggests new possible approaches aimed at the implementation of biosecurity measures against Campylobacter, considering the fact there are no successful control plans against it. The limit of this approach, already proposed in other contest, is represented by the difficulty of standardization as there are many variables affecting the characteristics of the bacterium, from the physical to the metabolic and pathogenic ability, as well as the farming conditions of birds. However, if associated to other measures could be useful in application of corrective actions applied in the meat chicken production chain addressed to minimize prevalence Campylobacter. The authors did not refer to the phenotypic and genotypic characteristics of antibiotic resistance of Campylobacter often related to morphological characteristics, i.e., biofilm, in some microorganisms. Nowadays, the antimicrobial resistance of Campylobacter spp. is considered more “insidious” than its effective zoonotic aspect.

Thank you for this comment. We added a paragraph on the phenotypic and genotypic characteristics of antibiotic resistance of C. jejuni.

  1. Finally, you should consider, if you like, an interesting work of Rawson et al (2020) as possible reference, for your introduction or discussion. Reference 10.3389/fmicb.2020.576646

Thank you, we also found this work very interesting and referenced it in the epidemiological studies section.

  1. Use italics for Campylobacter jejuni fig 1. Fig 3 line 268

This has been corrected.

  1. Line 172-173 I would omit this sentence; it is generally used when the Authors are directly responsible for the study.

Sorry, but we could not find the sentence that fit this description. On lines 172-173 we found this sentence: “These common traits might be used to inform the development of control measures for Campylobacter jejuni contamination through biofilms which would be potentially applicable for a wide range of C. jejuni strains.”, and we didn’t think that is the one you meant. We are however happy for the appropriate sentence to be removed.

  1. Line 231-232 “Instead, physical treatments involving temperature or water are mostly applied in that region”. Which region are you referring to? Please clarify sentence

We meant the European Union (EU). We realised that we did not include the explanation of this abbreviation – thank you for catching it. We also changed the word ‘region’ for ‘geographical area’

  1. line 398 per os or orally

This has been corrected.